# Predicting Short-Term Survival after Gross Total or Near Total Resection in Glioblastomas by Machine Learning-Based Radiomic Analysis of Preoperative MRI

**DOI:** 10.3390/cancers13205047

**Published:** 2021-10-09

**Authors:** Santiago Cepeda, Angel Pérez-Nuñez, Sergio García-García, Daniel García-Pérez, Ignacio Arrese, Luis Jiménez-Roldán, Manuel García-Galindo, Pedro González, María Velasco-Casares, Tomas Zamora, Rosario Sarabia

**Affiliations:** 1Department of Neurosurgery, University Hospital Río Hortega, 47012 Valladolid, Spain; segarciagarc@saludcastillayleon.es (S.G.-G.); iarreser@saludcastillayleon.es (I.A.); rsarabia@saludcastillayleon.es (R.S.); 2Department of Neurosurgery, University Hospital 12 de Octubre, 28041 Madrid, Spain; angelpnu@yahoo.es (A.P.-N.); daniel.garcia9@um.es (D.G.-P.); luisjroldan@hotmail.com (L.J.-R.); pedro.gleon@gmail.com (P.G.); 3School of Medicine, University of Valladolid (UVA), 47002 Valladolid, Spain; n42gagam@gmail.com; 4Department of Radiology, University Hospital Río Hortega, 47012 Valladolid, Spain; mvelascoca@saludcastillayleon.es; 5Department of Pathology, University Hospital Río Hortega, 47012 Valladolid, Spain; tzamorama@saludcastillayleon.es

**Keywords:** glioblastoma, radiomics, texture analysis, survival, machine learning

## Abstract

**Simple Summary:**

Identifying GBM patients with very short survival could contribute to adapting the therapeutic approach. According to our results, high-precision models can be elaborated using basic MRI sequences available at any center, combined with advanced image analysis. Although there are several previous publications related to this topic, a survival threshold that may be clinically relevant has not been proposed. The importance of our study lies in selecting patients with total or near-total resection, the short-term survival end-point applied of six months, and the employment of user-friendly software that allows clinicians to explore new statistical methodologies and carry out complex tasks such as the extraction of radiomic features. Promoting the use of these technological tools will motivate other clinical researchers to get involved and take advantage of radiomics and artificial intelligence, tools that have come to reinforce our analytical capacity.

**Abstract:**

Radiomics, in combination with artificial intelligence, has emerged as a powerful tool for the development of predictive models in neuro-oncology. Our study aims to find an answer to a clinically relevant question: is there a radiomic profile that can identify glioblastoma (GBM) patients with short-term survival after complete tumor resection? A retrospective study of GBM patients who underwent surgery was conducted in two institutions between January 2019 and January 2020, along with cases from public databases. Cases with gross total or near total tumor resection were included. Preoperative structural multiparametric magnetic resonance imaging (mpMRI) sequences were pre-processed, and a total of 15,720 radiomic features were extracted. After feature reduction, machine learning-based classifiers were used to predict early mortality (<6 months). Additionally, a survival analysis was performed using the random survival forest (RSF) algorithm. A total of 203 patients were enrolled in this study. In the classification task, the naive Bayes classifier obtained the best results in the test data set, with an area under the curve (AUC) of 0.769 and classification accuracy of 80%. The RSF model allowed the stratification of patients into low- and high-risk groups. In the test data set, this model obtained values of C-Index = 0.61, IBS = 0.123 and integrated AUC at six months of 0.761. In this study, we developed a reliable predictive model of short-term survival in GBM by applying open-source and user-friendly computational means. These new tools will assist clinicians in adapting our therapeutic approach considering individual patient characteristics.

## 1. Introduction

Glioblastoma (GBM) continues to be the most threatening primary brain neoplasm, with a median survival of approximately 15 months [1]. Currently, despite the standard treatment that includes maximum safe surgical resection followed by adjuvant chemoradiation therapy [2,3], its prognosis remains ominous, and our knowledge of this neoplasm is still limited.

Predicting a patient’s survival is of vital importance for determining the ideal choice of treatment and management. Currently, several prognostic factors are commonly used to predict the prognosis of these patients, including age, sex, Karnofsky performance status (KPS), molecular profile, extent of resection, preoperative tumor volume, volume of non-enhancing tumor and degree of necrosis [4]. However, some of these features depend on radiologists’ interpretation, which justifies the increasing need for an unbiased and quantitative radiological evaluation.

Magnetic resonance imaging (MRI) plays a fundamental role in neuro-oncology for diagnosing and assessing response to treatment and is being increasingly used as a non-invasive predictive tool. On the other hand, the term "radiomics" refers to the process of obtaining quantitative features based on the intensity, volume, shape and texture variations of the radiological images and creating algorithms that find the association of these variables with the survival and outcome of the patients [5]. Through radiomics, converting medical images into high-dimensional data allows us to expose the underlying pathophysiology, especially intratumor heterogeneity [6]. This extraction process captures tumor characteristics undetectable to the human eye and gives added value to clinical visual perception. Radiomics incorporate several essential disciplines, including radiology for image interpretation, computerized vision for extracting quantitative variables, and machine learning for classification and regression tasks [7]. Such integration has been demonstrated to exceed expert human abilities in multiple tasks, including diagnosis and outcome prediction.

Recognizing patients who would not benefit from standard treatment and identifying those who need a more aggressive approach at the time of diagnosis is essential for managing GBM through personalized medicine [8]. There are several publications that, through the integration of radiomics and artificial intelligence, seek to establish survival prediction models in GBM based on preoperative MRI [8,9,10,11,12,13,14]. In the vast majority of studies, patients are classified according to their survival into two or three categories, depending on whether they exceed 10 or 15 months of survival. This approach aims to identify medium- and long-term survivors who could theoretically be subsidiaries of aggressive therapies. Furthermore, in most studies, the extent of resection is not used as a discriminatory factor, and biopsies and partial and subtotal resections are included.

This fact precludes the implementation of such predictive models in newly diagnosed GBM patients. Our study aims to use the radiological characteristics from structural preoperative multiparametric magnetic resonance imaging (mpMRI) to construct a predictive model of short-term survival in patients in whom total or near-total resection of the enhancing tumor has been performed followed by standard treatment.

## 2. Materials and Methods

### 2.1. Study Population

A retrospective collection of patients who underwent surgery with a diagnosis of GBM was carried out in two institutions between January 2019 and January 2020. In addition, a second cohort of patients was selected from available public databases: the BraTS (Multimodal Brain Tumor Segmentation) Challenge 2020 [15,16,17], and three other sources available through The Cancer Image Archive [Ivy Glioblastoma Altas Project (Ivy -GAP) [18], the National Cancer Institute’s Clinical Proteomic Tumor Analysis Consortium Glioblastoma Multiforme (CPTAC-GBM) and The Cancer Genome Atlas Glioblastoma Multiforme (TCGA-GBM) [19]]. The inclusion criteria were pathologically confirmed glioblastomas, availability of preoperative MRI with structural/conventional sequences [T2-weighted images (T2WI), fluid-attenuated inversion recovery (FLAIR), T1-weighted images (T1WI) and contrast-enhanced T1-weighted images (T1CE)] with adequate resolution, known survival status, and clinical information (age and type of surgical resection). Only those cases in which gross total resection (100% of the enhancing tumor volume) or near-total resection (>95% of the enhancing tumor volume) were included. Patients were randomly allocated into training and testing data sets following a proportion of 70/30. The primary endpoint was overall survival (OS), which was defined as the number of days from the initial pathological diagnosis to death (censored = 1) or the last date that they were known to be alive (censored = 0). Public data sets do not have patient identifiers, hence, no institutional review board approval was required. Nevertheless, the study was approved by the institutional review boards and ethics committees of the other two participating centers. Additionally, all institutional patients provided written informed consent. The study was performed in accordance with the ethical standards as laid down in the 1964 Declaration of Helsinki and its later amendments.

### 2.2. Image Data Description and Preprocessing

All BraTS scans were acquired with different clinical protocols and various scanners from multiple (*n* = 19) institutions. Details of the protocol acquisition of the scans from TCIA and institutional cases are shown in Appendix A.

Image pre-processing consists of several steps. First, mpMRI scans were converted to Neuroimaging Informatics Technology Initiative (NifTI) format. Then, the scans were placed in a common orientation [“LPS” (left-posterior-superior) in the radiological convention or “RAI” (right-anterior-inferior” in the neurological convention)]. Later, the scans for every subject were registered to SRI24 anatomical atlas space [20]. N4 bias correction [21] was applied as a temporary step to facilitate optimal registration but was not included at the end of the process since it might obliterate the MRI signal, particularly on the FLAIR modality [17].

The T1W1, T2WI and FLAIR scans were registered to the transformed T1CE scan, resulting in coregistered resampled volumes of 1 × 1 × 1 mm isotropic voxels. The brain was then extracted from all co-registered scans using a pretrained deep learning-based model [22]. Finally, intensity Z-scoring normalization was carried out. All pre-processing pipelines were generated using The Cancer Imaging Phenomics Toolkit (CaPTk) [23].

### 2.3. Tumor Segmentation and Feature Extraction

The method used to generate the segmentation labels of the different tumor regions is called GLISTRboost (Boosted GLioma Image SegmenTation and Registration) [24], which is defined as a hybrid generative-discriminative tumor segmentation method. This segmentation algorithm comprises a glioma growth model, a discriminative part based on a gradient boosting multiclass classification scheme and a Bayesian strategy [14]. Segmentation labels or volumes of interest (VOIs) were as follows: enhancing tumor (ET), non-enhancing tumur/necrosis (NET), and edema (ED). Segmentations were evaluated by two experts (S.C., S.G.-G.) and corrected manually if necessary.

Using the extraction tool of CaPTk, a total of 15,720 characteristics were computed from the tumor subregions (i.e., ET, NET and ED) and the four mpMRI modalities following the Image biomarker standardization initiative (IBSI) [25] definitions. These extracted features included intensity features or first-order statistics, histogram-based features, and volumetric, morphologic and textural features, including those based on the gray level cooccurrence matrix (GLCM), gray level run-length matrix (GLRLM), neighborhood gray-tone difference matrix (NGTDM), gray level size-zone matrix (GLSZM) and lattice-based features. A detailed description of these characteristics is shown in Appendix A.

### 2.4. Data Processing and Feature Selection

After the extraction of the radiomic features, the data were pre-processed by data cleaning (removing the variables with more than 5% missing values) and imputation (using the average value method). Afterward, 13,265 features were z-score normalized based on the mean and standard deviation. Then, a feature selection process was necessary to reduce these high-dimensional imaging features to avoid overfitting. Feature selection helps to optimize the generalizability and reproducibility of the models subsequently built. The feature selection process was performed using the training data set exclusively. A two-step selection method was used as follows. Spearman’s correlation coefficient was calculated for each pair of radiomic features. Then, the features with Spearman’s correlation coefficient > 0.95 with each other were discarded, retaining a single feature in each set. Later, features were reduced by including only the variables with a significance <0.001 with the OS in days. Thus, the number of features was reduced to a set of 260.

### 2.5. Statistical Analysis and Machine Learning

Predicting OS was achieved by two different strategies. The first is a binary classification task between short and long survival. For this purpose and after feature reduction, several classification algorithms were assessed for patient stratification. As a previous step, machine learning (ML)-based filters were used: Gini index (GINI), fast correlation-based filter (FCBF) and information gain (InfoGain). Hence, the top 10 features were selected. Then, six different ML classifiers were trained: logistic regression (LR), naive Bayes, k-nearest neighbors (kNN), random forest (RF), support vector machine (SVM) and a multilayer perceptron algorithm (neural network -NN). We used the default settings of the hyperparameters detailed in the Appendix A. The target response for each model was the patient’s OS grouped into two classes to distinguish patients who survived <6 months (short-term survivors) from others. Then, the results were quantitatively validated on the testing data set. The performance of the ML classifiers was measured by the area under the receiver operating characteristic curve (AUC), accuracy, precision, F1 score and recall. All performance metrics were reported as the average value over classes.

The second statistical strategy was conducted using the random survival forest (RSF) approach from the R package “randomForestRSC” [26], an ensemble-tree method that adapts random forests to right-censored data and survival analysis. RSF does not rely on restrictive assumptions such as proportional hazards and automatically handles nonlinear effects and interactions of high-dimensional data. Features were ranked by positive importance using a variable hunting algorithm as a feature selector. Model hyperparameters were as follows: number of trees = 500, node size = 2, number of splits = 10 and log-rank as the splitting rule. Training predictions were performed using 5-fold cross-validation. We also evaluated the model’s ability to generalize those predictions on the testing group. 

When the primary outcome is survival (time to event), RSF produces a cumulative hazard function (CHF) from each decision tree that is averaged in an ensemble out-of-bag CHF (OOB-CHF). The predicted ensemble mortality is the mean OOB-CHF estimated by the RSF model for each subject, and it was used to calculate each patient’s estimated mortality risk. We used the results from the RSF model to build a mortality risk score and split the sample into high- and low-risk groups. The OOB-CHF cut-off values defining the risk groups were calculated through the “cutp” function of the “survMisc” package [27]. The log-rank test was used to compare the survival Kaplan–Meier curves between the patients in the high- and low-risk groups.

Finally, Cox proportional hazard regression models were fitted to the training data set using the dichotomized risk score (high- and low-risk groups) from the RSF model as an explanatory variable, the patient’s age and a combination of both. Then, the models were validated in the testing data set. The performance assessment of the survival models was performed by calculating the prediction errors using the integrated Brier score (IBS) defined as the average squared distances between the observed survival status and the predicted survival probability by the “pec” package [28]. Additionally, the discriminatory capacity of the model was evaluated by calculating the concordance index (CI), which refers to the probability that, for a pair of randomly allowed samples, the sample with the highest risk prediction experiences an event before the sample with the smallest risk. Furthermore, the integrated area under the time-dependent ROC curve (iAUC) was calculated for all models using the “risksetROC” package [29]. The standard approach of ROC curve analysis considers event (death) status and predictor value for an individual as fixed over time. Because the status and explanatory variables change over time, we used the risksetROC package that estimates the iAUC under incident sensitivity and dynamic specificity definition and produces accuracy measures for censored data under proportional or nonproportional hazard assumption of Cox regression estimator [30]. Following the objective of our study, we also calculated the iAUC at six months for all models.

Statistical and survival analyses were performed with R version 4.0.5 (R Foundation for Statistical Computing, Vienna, Austria). The differences in age, OS, the proportion of right-censored cases and short-term (<6 months) survival cases were assessed using Student’s t-test, Mann–Whitney U test and two-proportions Z-test, respectively. For the binary classification model, we used Orange version 3.28.0 (University of Ljubljana, Ljubljana, Slovenia) [31]. The radiomics quality score (RQS) was calculated for this study according to the recommendations by Lambin et al. [32]. A *p* value < 0.05 was considered to indicate a statistically significant difference. The image processing and statistical analysis workflow are shown in Figure 1.

## 3. Results

### 3.1. Patient Population

Two hundred and three patients were enrolled in this study. The mean age was 61.49 ± 11.76 (range 27.81–86.65). The median OS was 407 days [interquartile range (IQR) = 351.5]. A total of 7.9% (16) of patients were right-censored cases, and 17.24% (35) registered an OS of less than six months. The patients were randomly assigned to a training data set of 143 patients and a testing data set of 60 patients. There were no significant differences in age, preoperative tumor volume, OS, or proportion of right-censored and short-term survival cases between the training and test data sets (Table 1).

### 3.2. Classification Task and Survival Groups

The variable selection filters made it possible to determine the top ten radiomic features (Table 2). Based on these characteristics and using the ML classifiers, patients were classified into short-term survivors (<6 months).

The optimal results were obtained by applying the information gain as a feature selector. Thus, in the training cohort, AUC values were achieved with a range between 0.802 and 0.978, a classification accuracy between 81.8% and 94.4%, and a precision between 82.8% and 94.8% (Appendix A).

In the testing data set, the naive Bayes classifier obtained the best results, with an AUC of 0.769, a classification accuracy of 80%, and a precision of 81% (Table 3 and Figure 2).

### 3.3. Random Survival Forest to Predict OS

In the radiomic model based on RSF, the variable-hunting algorithm selected 17 radiomic features to predict OS in the training data set (Table 4).

Based on these characteristics, the mortality risk score was calculated using the OOB-CHF. The cut-off point used was 0.684 (Appendix A). This cut-off point allowed patients to be stratified into low-risk and high-risk groups [HR = 2.19, (95% CI: 1.54–3.12), log-rank *p* = < 0.001, C-Index = 0.61, IBS = 0.096]. In the testing data set, patients were also stratified using the same cut-off point [HR 2.16, (95% CI: 1.21–3.89), log-rank *p* = 0.008, C-Index = 0.61, IBS = 0.123] (Figure 3). The multivariate Cox regression models in which age was incorporated as an explanatory variable are shown in Table 5.

The iAUC of the radiomic model was 0.591 in the training data set and 0.568 in the test data set. By incorporating age as a variable in the model, the iAUC increased to 0.650 in the training data set and 0.627 in the testing data set.

After setting the survival time to 6 months, the predictive accuracy of the radiomic model improved to an iAUC of 0.712 in the training data set and 0.761 in the testing data set (Appendix A).

The RQS was used to evaluate the methodological quality of our study. We obtained a score of 19/36 (53%). A detailed report of RQS items is shown in Appendix A.

## 4. Discussion

In the present study, we elaborated a prediction model of short-term survival with high predictive capacity using the radiomic features of structural preoperative multiparametric MRI of GBM patients.

We believe that the main strength of our study is based on a selection of patients who underwent total or near-total resection of the enhancing tumor. We considered this methodologic aspect due to the undeniable link between the extent of resection and survival in these patients [33]. In most previous studies, the extent of resection was not used as a selection criterion, including partial resections and biopsies in their series, without making any adjustment during the analysis phase. The exception is the studies by Bakas et al. [14] and Fathi et al. [34], in which the entire cohort of patients underwent complete resection and standard chemoradiotherapy treatment.

Another crucial point of our work is to set our objective to identify short-term survival patients, in contrast to previously published studies where 10 and 12–15 months were used as cut-off points for defining short- and long-term survival, respectively [9,10,12,35,36,37,38,39]. The only reference we found is in the work of Prasanna et al. [40], who classified patients in long (>18 months) versus short-term (<7 months) survival, based on peritumoral region radiomic features. The rationale of our approach lies in the desire to predict the survival of patients diagnosed with GBM by non-invasive methods and to identify those with very short survival. In these patients, the futility of our treatments would lead us to offer patients and their families the option of not taking aggressive measures or, on the contrary, opening new lines of research since those cases would be poor responders to the standard therapies applied currently.

As another strength of our work, we can mention the use of open-source software. The CapTk and Orange programs have a very intuitive yet robust user interface, thanks to which clinicians can access advanced image processing technics and data mining tools. Thanks to these programs, we have performed complex tasks such as automatic tumor segmentation, image processing, radiomic feature extraction, and exploring different ML-based algorithms.

Concerning statistical analysis, we used a dual approach. On the one hand, we have used a binary classification system using different ML-based algorithms. Additionally, we used state-of-the-art survival analysis techniques such as random survival forest and time-dependent ROC curve analysis focused on short-term survival that contribute to corroborating the stability of the models produced here.

We also highlight that the results of our predictive models have been achieved using only structural MRI [14]. These results could even be improved after the inclusion of studies based on diffusion and perfusion sequences [41]. However, basic MRI is available in most centers, and according to our results, the lack of special sequences is not a limitation in the search for useful radiological patterns in clinical practice.

An important aspect to discuss is the biological correspondence of the variables employed by the prediction models. There is notable variability concerning the radiomic characteristics used by previous studies, which is one of the most significant obstacles in reproducing and validating their results. In our study, most of the selected variables came from the T1CE sequence followed by FLAIR and T2WI, while the different tumor subregions (i.e., ET, NET and ED) were represented in the models in a balanced way. In our series, first-order features and morphological characteristics appeared to be important for OS prediction.

We are aware of the limitations of our work, such as the lack of clinical and molecular data that can be incorporated into predictive models. Even so, age as an explanatory variable was incorporated into our models due to its significant association with the OS of these patients, proving that its mere incorporation into the analysis improved the performance of the models. Despite having a relatively small sample size, various statistical techniques have been applied to overcome the "curse of dimensionality". Taking into consideration that MRI studies come from numerous sources, the processing method for image standardization that we have chosen aims to be simple and reliable and has been used by several studies [23,34,42].

Another limitation of predictive models based on radiomic features is the lack of assessment of the repeatability and reproducibility of the features calculated from the magnetic resonance images. We have included a detailed description of the feature extraction software, digital image manipulation, and image acquisition parameters to improve reporting quality. However, it is necessary to carry out new studies specifically designed to solve this problem.

Unquestionably, the combination of texture analysis and artificial intelligence is starting to facilitate knowledge about the biological behaviour of GBMs through the study of their patient-dependent heterogeneity. However, the rapid development of big data tools and the tremendous complexity of advanced medical image analysis dangerously threaten widening the gap between data experts and clinicians. Then, it is a paradox that radiomics, defined by Lambin et al. [32] as “the bridge between medical imaging and personalized medicine”, is now out of reach of those who treat real patients every day. Therefore, our study arises from a real need and aims to find a solution to a clinically relevant problem: identifying GBM patients with short survival after complete resection. Although our results can be improved, we show that there are currently computer tools and public data sets available to everyone to develop reliable predictive models. Hence, our duty as clinicians is to become immersed in developing these models, since our pragmatism can never be replaced, even by the most complex algorithm.

Indeed, our results are encouraging, and the precision achieved is similar to the previous literature. However, this article represents an early age of a promising future in which the ultimate link between image, diagnosis and prognosis could finally be decoded to provide instant, useful and precise information to individual patients based on their specific features. Multi-institutional studies [43] would allow the generalization of predictive models, or even adapt the mechanisms of data pre-processing, extraction, and analysis to the MRI from each center, since the standardization of acquisition protocols is not feasible. Finally, we believe that in this catastrophic disease, the quality of life of our patients should be our first consideration, and maximum exploitation of available neuroimaging techniques should be pursued to optimize management strategies, avoiding unnecessarily aggressive therapies in those patients who will not benefit from them.

## 5. Conclusions

In the present study, we evaluated the capability of the radiomic features of preoperative mpMRI and machine learning-based classification and regression analysis to predict short-term survival in GBM patients. Our model shows a classification accuracy of 80% and an iAUC of 0.76 to predict OS < 6 months in the test data set. We believe that these new tools will serve clinicians to understand the biological behaviour of individual GBMs, and we must take advantage of them.

## Figures and Tables

**Figure 1 cancers-13-05047-f001:**
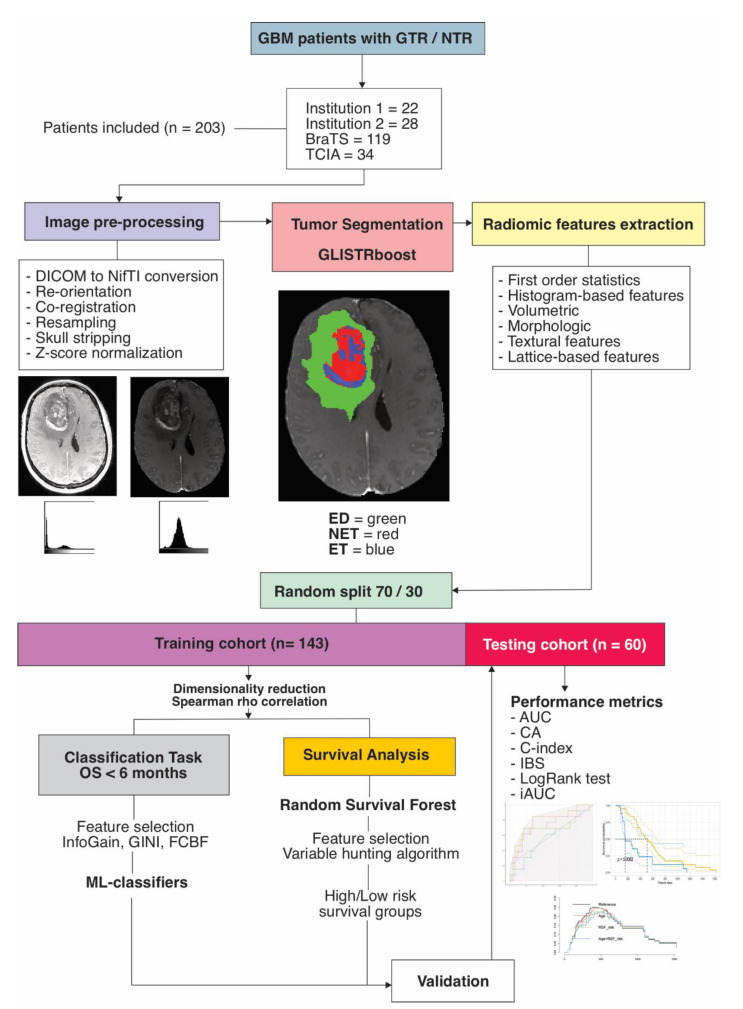
Study workflow. GBM = glioblastoma, GTR = gross-total resection, NTR = near-total resection, BraTS = Brain tumor segmentation challenge 2020, TCIA = The cancer image archive, GLISTRboost = Boosted GLioma Image SegmenTation and Registration, ED = edema, NET = nonenhancing tumor, ET = enhancing tumor, OS = overall survival, InfoGain = Information gain, GINI = Gini Index, FCBF = Fast correlation-based filter, ML = machine learning, AUC = area under the curve, CA = classification accuracy, IBS = integrated Brier score, iAUC = integrated AUC.

**Figure 2 cancers-13-05047-f002:**
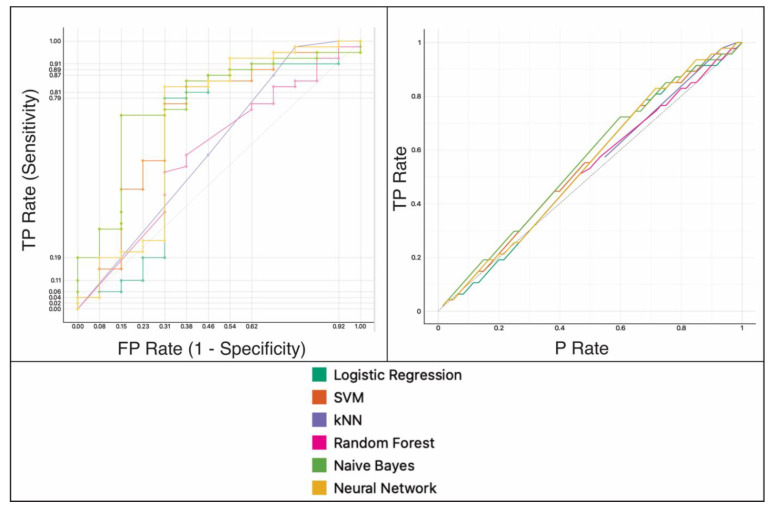
Performance assessment of the binary classification model on the testing data set. left: area under the receiver operating characteristic (ROC) curve and right: calibration plot. kNN = k-nearest neighbor, SVM = support vector machine, TP = true positives, FP = false positives, P = positives.

**Figure 3 cancers-13-05047-f003:**
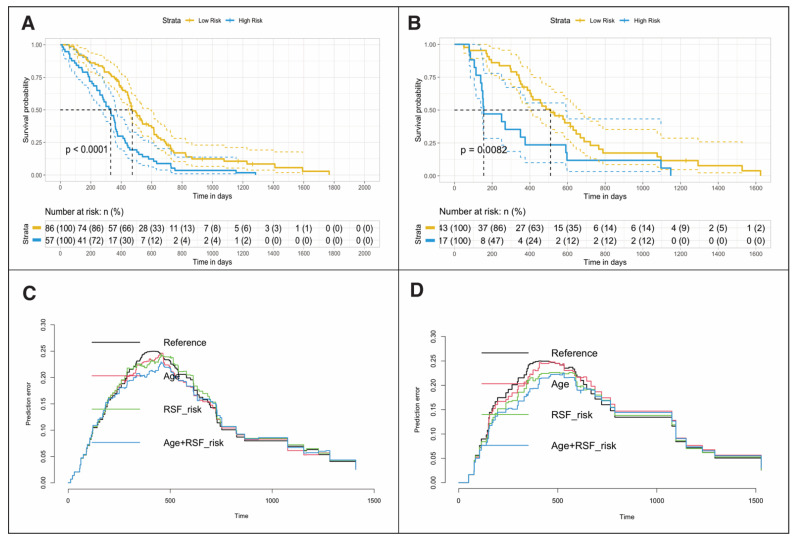
Kaplan–Meier plots showing differences in overall survival for patients in training (**A**) and testing (**B**) data sets stratified into low- or high-risk groups by random survival forest (RSF)-based ensemble mortality. Prediction error curves show Cox regression model performance in the training (**C**) and testing (**D**) groups. HR= Hazard Ratio, RSF = Random Survival Forest, IBS = Integrated Brier Score, iAUC = Integrated area under the curve, 6m – AUC = six months - area under the curve.

**Table 1 cancers-13-05047-t001:** Patient characteristics.

Data Set	n	Age (SD)	Tumor Volume (cm^3^)	OS (IQR)	Censored (%)	OS < 6 m (%)
BraTS	119	62 ± 12	30.09 ± 11.77	374 (364)	0	21.8% (26)
TCIA	34	60.3 ± 10	42.55 ± 15.71	414 (482)	5.9% (2)	17.6% (6)
Institution 1	22	65.3 ± 10	40.77 ± 18.12	451 (307)	22.7% (5)	13.6% (3)
Institution 2	28	57.9 ± 13	42.14 ± 17.16	466 (217)	32.1% (9)	0
After random split (70/30)						
Training	143	62.1 ± 11	40.62 ± 10.27	409 (311)	7.7% (11)	17.5% (25)
Testing	60	60.1 ± 13	39.27 ± 11.35	404 (392)	8.3% (5)	16.7% (10)
Statistical comparison between cohorts		*t* = 0.10, *p* = 0.273	*t* = 0.12, *p* = 0.902	*U* = 0.33, *p* = 0.743	χ^2^ = 0.02, *p* = 0.877	χ^2^ = 1.17, *p* = 0.279

BraTS = Brain Tumor Segmentation Challenge 2020, TCIA = The Cancer Image Archive, SD = standard deviation, IQR = interquartile range.

**Table 2 cancers-13-05047-t002:** Radiomic features ranked by different scoring methods for short-term survival prediction.

Filter		
Information Gain	Gini Index	FCBF
T1CE_ED_Lattice_Histogram_Bins-20_Radius-1_Bins-20_NinetyFifthPercentile_Skewness	T1CE_ET_Lattice_Histogram_Bins-20_Radius-1_Bins-20_Range_Max	T1CE_ED_Lattice_Histogram_Bins-20_Radius-1_Bins-20_NinetyFifthPercentile_Skewness
T1CE_ED_Lattice_Histogram_Bins-20_Radius-1_Bins-20_NinetyFifthPercentile_Kurtosis	T1CE_ED_Lattice_Histogram_Bins-20_Radius-1_Bins-20_NinetyFifthPercentile_Kurtosis	T2WI_ED_Lattice_Intensity_Bins-20_Radius-1_QuartileCoefficientOfVariation_StdDev
T1CE_ET_Lattice_Histogram_Bins-20_Radius-1_Bins-20_Range_Max	T1CE_ED_Lattice_Histogram_Bins-20_Radius-1_Bins-20_NinetyFifthPercentile_Skewness	T1CE_ET_Lattice_Histogram_Bins-20_Radius-1_Bins-20_Energy_Skewness
T2WI_ED_Lattice_Intensity_Bins-20_Radius-1_QuartileCoefficientOfVariation_StdDev	FLAIR_ET_Lattice_Morphologic_EquivalentSphericalRadius_Variance	T1CE_ED_Lattice_Histogram_Bins-20_Radius-1_Bins-20_NinetyFifthPercentile_Kurtosis
FLAIR_ET_Lattice_Morphologic_EquivalentSphericalPerimeter_Variance	FLAIR_ET_Lattice_Morphologic_EquivalentSphericalPerimeter_Variance	T1CE_ET_Lattice_Histogram_Bins-20_Radius-1_Bins-20_Range_Max
FLAIR_ET_Lattice_Morphologic_EquivalentSphericalRadius_Variance	FLAIR_NET_Lattice_Histogram_Bins-20_Bins-20_Bin-12_Frequency_Median	FLAIR_ET_Lattice_Morphologic_EquivalentSphericalPerimeter_Variance
FLAIR_NET_Lattice_Histogram_Bins-20_Radius-1_Bins-20_InterQuartileRange_StdDev	T2WI_ED_Lattice_Intensity_Bins-20_Radius-1_Mean_Skewness	FLAIR_ET_Lattice_Morphologic_EquivalentSphericalRadius_Variance
FLAIR_NET_Lattice_Histogram_Bins-20_Radius-1_Bins-20_InterQuartileRange_Variance	T2WI_ED_Lattice_Intensity_Bins-20_Radius-1_QuartileCoefficientOfVariation_StdDev	FLAIR_NET_Lattice_Histogram_Bins-20_Radius-1_Bins-20_InterQuartileRange_StdDev
FLAIR_NET_Lattice_Histogram_Bins-20_Bins-20_Uniformity_Min	FLAIR_NET_Lattice_Histogram_Bins-20_Bins-20_Uniformity_Min	FLAIR_NET_Lattice_Histogram_Bins-20_Bins-20_Bin-12_Frequency_Median
A10824 FLAIR_NET_Lattice_Histogram_Bins-20_Bins-20_Bin-12_Frequency_Median	FLAIR_NET_Lattice_Histogram_Bins-20_Radius-1_Bins-20_InterQuartileRange_Variance	FLAIR_NET_Lattice_Histogram_Bins-20_Radius-1_Bins-20_InterQuartileRange_Variance

FCBF, fast correlation-based filter, T1CE = contrast-enhanced T1-weighted images, T2WI = T2 weighted images, FLAIR = Fluid-attenuated inversion recovery, ET = enhancing tumor, NET = non-enhancing tumor, ED = edema.

**Table 3 cancers-13-05047-t003:** Model performance grouped by feature selection filter and machine learning classifier on the testing data set.

Classifier	Filter	AUC	CA	Precision	F1
Naive Bayes	Information Gain	0.769	0.800	0.812	0.805
	Gini Index	0.784	0.767	0.810	0.781
	FCBF	0.743	0.783	0.803	0.791
k-Nearest Neighbor	Information Gain	0.600	0.817	0.806	0.776
	Gini Index	0.639	0.800	0.771	0.763
	FCBF	0.670	0.767	0.713	0.724
Neural Network	Information Gain	0.691	0.800	0.771	0.763
	Gini Index	0.682	0.767	0.691	0.705
	FCBF	0.722	0.817	0.806	0.776
Random Forest	Information Gain	0.574	0.733	0.683	0.701
	Gini Index	0.666	0.750	0.696	0.713
	FCBF	0.700	0.783	0.738	0.735
Support Vector Machine	Information Gain	0.709	0.750	0.608	0.671
	Gini Index	0.630	0.750	0.608	0.671
	FCBF	0.730	0.800	0.777	0.747
Logistic Regression	Information Gain	0.648	0.783	0.614	0.688
	Gini Index	0.643	0.783	0.614	0.688
	FCBF	0.656	0.783	0.614	0.688

AUC: area under the curve; CA: classification accuracy; FCBC: fast correlation-based filter.

**Table 4 cancers-13-05047-t004:** Radiomic features selected by the variable hunting algorithm for Random Forest Survival prediction.

Nº	**Feature**
1	T1CE_NET_Lattice_GLSZM_Bins-20_Radius-1_LargeZoneHighGreyLevelEmphasis_Mean
2	FLAIR_ED_Lattice_NGTDM_Busyness_Max
3	T1CE_NET_Lattice_GLSZM_Bins-20_Radius-1_LargeZoneHighGreyLevelEmphasis_Median
4	FLAIR_NET_Lattice_Histogram_Bins-20_Radius-1_Bins-20_Bin-14_Frequency_Max
5	FLAIR_NET_Lattice_Intensity_Bins-20_Radius-1_NinetiethPercentile_Mean
6	T1CE_ET_Lattice_Intensity_Bins-20_Radius-1_StandardDeviation_StdDev
7	FLAIR_NET_Lattice_Morphologic_PixelsOnBorder_Variance
8	T1CE_ET_Lattice_Histogram_Bins-20_Radius-1_Bins-20_Bin-0_Probability_Kurtosis
9	T1CE_ET_Lattice_Histogram_Bins-20_Radius-1_Bins-20_Bin-9_Probability_Median
10	T1CE_ET_Lattice_GLSZM_Bins-20_Radius-1_ZoneSizeMean_Variance
11	T2WI_ED_Lattice_Intensity_Bins-20_Radius-1_Mean_Skewness
12	FLAIR_ET_Lattice_Morphologic_Perimeter_Skewness
13	FLAIR_NET_Lattice_Histogram_Bins-20_Radius-1_Bins-20_FifthPercentileMean_Max
14	FLAIR_ET_Lattice_Morphologic_EquivalentSphericalRadius_Variance
15	T1CE_ET_Lattice_GLRLM_Bins-20_Radius-1_RunLengthNonuniformity_Kurtosis
16	FLAIR_ED_Lattice_Intensity_Bins-20_Radius-1_InterQuartileRange_Median
17	FLAIR_ED_Lattice_Histogram_Bins-20_Radius-1_Bins-20_Sum_Max

T1CE = contrast-enhanced T1-weighted images, T2WI = T2 weighted images, FLAIR = Fluid-attenuated inversion recovery, ET = enhancing tumor, NET = non-enhancing tumor, ED = edema, GLSZM = Gray Level Size-Zone Matrix, NGTDM = Neighborhood Gray-Tone Difference Matrix, GLRLM = Gray Level Run-Length Matrix.

**Table 5 cancers-13-05047-t005:** Univariate and Multivariate Cox regression analysis.

Univariate Cox Regression Analysis								
Variable	β	HR	95% CI	*p*	C-Index	IBS	iAUC	6m-iAUC
Training data set								
Age	0.03	1.03	1.02–1.05	<0.001	0.61	0.089	0.604	0.599
Radiomic RSF Score (High risk)	0.78	2.19	1.54–3.12	<0.001	0.61	0.096	0.591	0.712
Testing data set								
Age	0.03	1.03	1–1.06	0.023	0.60	0.128	0.592	0.643
Radiomic RSF Score (High risk)	0.77	2.16	1.21–3.89	0.009	0.61	0.123	0.568	0.761
**Multivariate Cox Regression Model**								
**Model**	**Likelihood Ratio Test**				**C-Index**	**IBS**	**iAUC**	**6m-AUC**
	**χ** ** ^2^ **		**df**	** *p* **				
Training data set								
Age + Radiomic RSF Score (High risk)	36.93		2	<0.001	0.66	0.084	0.650	0.730
Testing data set								
Age + Radiomic RSF Score (High risk)	11.35		2	0.003	0.68	0.118	0.6278	0.769

## Data Availability

The data sets generated during and/or analysed during the current study are available from the corresponding author on reasonable request.

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
