# Peer review of "Predicting Short-Term Survival after Gross Total or Near Total Resection in Glioblastomas by Machine Learning-Based Radiomic Analysis of Preoperative MRI"

_cancers, 2021, doi:10.3390/cancers13205047_

Round 1
Reviewer 1 Report
In this manuscript, the authors investigated the feasibility of radiomics analysis of preoperative MRI for predicting short-term survival after gross total or near total resection in glioblastomas.
Major Strengths;
--- This study was aimed to the patients with gross total or near total resection
--- Rigorous data analysis process with multi-institution validation
Major Weaknesses;
--- The basic patient characteristics, such as gender and tumor size, should be provided in detail.
--- why random split was used in this study. Two datasets of them could be selected for model training and the remaining two could be used for independent validation.
--- Different sequences were not analyzed individually. The best MRI sequence for predicting OS deserved to be explored.
--- As a commonly used classifier, the classification results of logistic regression was suggested to be added.
--- Given that Age and Radiomic RSF Score achieved a similar performance with C-index of 0.61, stepwise cox regression analysis should be adopted to explore that whether radiomics score have the supplementary predictive value.
Author Response
REVIEWER 1:
In this manuscript, the authors investigated the feasibility of radiomics analysis of preoperative MRI for predicting short-term survival after gross total or near total resection in glioblastomas.
Major Strengths;
--- This study was aimed to the patients with gross total or near total resection
--- Rigorous data analysis process with multi-institution validation
Major Weaknesses;
--- The basic patient characteristics, such as gender and tumor size, should be provided in detail.
We have added in Table 1 the description of the tumor volume for each data set and the training and test groups. Unfortunately, sex is a missing variable in the public data set. For this reason, we have not been able to include it even though we acknowledge its relevance.
--- why random split was used in this study. Two datasets of them could be selected for model training and the remaining two could be used for independent validation.
Indeed, a valid option for this type of analysis is to use a data set for training the model and save one or more groups for testing. However, we observed significant differences between data sources. For example, Table 1 shows that the overall survival values are lower in the BraTS dataset compared to the other groups. Furthermore, the proportion of censored cases and cases with short-term survival also has an uneven distribution among the datasets. For this reason, we preferred to perform a random division of the sample and thus balance the distribution of the variables of interest. As a result, there are no significant differences in these variables between the training and test groups (line 239).
--- Different sequences were not analyzed individually. The best MRI sequence for predicting OS deserved to be explored.
Our prediction model is designed to take advantage of all the basic MRI sequences. When we developed the extraction pipeline and then the data handling, we believed the best option was to include all the multiparametric data. Thus, the correlations between the features of each sequence are not ignored. In addition, individualized analysis of each MRI sequence and each tumor region would require a bigger study sample. As a result, our predictive model is not based on a single sequence, and it would be difficult to determine which of them is the most important. Instead, our model combines intensity, morphological and texture characteristics from the three tumor regions and the different MRI sequences. This way of taking advantage of the information from clinical images is similar to what we do in our clinical practice.
--- As a commonly used classifier, the classification results of logistic regression was suggested to be added.
Following your suggestion, we have added logistic regression as a classifier. The results are shown in Table 3, Figure 2 and Supplementary Table S4.
--- Given that Age and Radiomic RSF Score achieved a similar performance with C-index of 0.61, stepwise cox regression analysis should be adopted to explore that whether radiomics score have the supplementary predictive value.
Indeed, the C-Index values are similar for the variable age and RSF-score. By the way, one of the alternatives we tested during the analysis phase was to use the stepwise Cox regression, using the My.stepwise.coxph function from the My.stepwise R package. As a result, we obtained that the two variables were retained in the final model.
For this reason, to demonstrate their individual importance, we show in Table 5 Cox univariate regressions with their respective values of beta coefficient, HR and p-values. More importantly, the IBS and the iAUC-6m show a slight difference in favor of the RSF-score.
Reviewer 2 Report
It is of clinical value to predict OS of GBM after resection. The method of machine learning on preoperative MR images radiomics is sound, the paper is also well-written. For this type of study, the data are typically separated as training, validation and test sets. It seems the testing data in this study is actually a validation set (line 180-181). if there is no test data for independent testing, this would limit the results of the study.
The paper uses various models that need some pre-defined hyperparameters. How are these parameters defined? If they are derived by maximizing model performances, the calculated model performances such as AUCs are biased. Likewise, initial feature selection by correlation with OS (Line 156) on whole data would also lead to information leakage to test data, so to bias the performance evaluation.
Similar to many radiomics studies, this study is limited by the small number of data samples. Radiomics create thousands of features, meanwhile, the modeling, especially the neural network, uses many parameters. All these require a great number of data samples for training and testing. data samples 143/60 may be not enough.
Another common limitation for radiomics is the potential lack of reproducibility and reliability of radiomics feature calculations. While this could be out of scope of this study, the study involves MR from multiple sources. Could the authors evaluate the differential power of a feature between different data sources, and discuss the consistency of the feature predictive power across all the data sources.
Author Response
REVIEWER 2:
It is of clinical value to predict OS of GBM after resection. The method of machine learning on preoperative MR images radiomics is sound, the paper is also well-written. For this type of study, the data are typically separated as training, validation and test sets. It seems the testing data in this study is actually a validation set (line 180-181). if there is no test data for independent testing, this would limit the results of the study.
In the Methods section and Figure 1, we stated that the total of cases was randomly divided into a training group used for learning and fit the model and a test group used only to provide an unbiased evaluation of the models (line 102). Throughout the manuscript, we have made corrections to clarify that it is a test group and not a validation data set. Moreover, on line 182, we have corrected: "Training predictions were performed using 5-fold cross-validation. We also evaluated the model’s ability to generalize those predictions on the testing group. "
The paper uses various models that need some pre-defined hyperparameters. How are these parameters defined? If they are derived by maximizing model performances, the calculated model performances such as AUCs are biased. Likewise, initial feature selection by correlation with OS (Line 156) on whole data would also lead to information leakage to test data, so to bias the performance evaluation.
We have used the default configuration of the hyperparameters offered by the software (Orange) for the ML-classifiers. However, following your suggestion, we have added a detailed description in the Method section (line 168) and Supplementary Table S3.
Regarding dimensionality reduction and feature selection, we have added a clarification: "The feature process was performed using the training data set exclusively."(line 153) Again, we emphasize that in Figure 1, we had already correctly noted this step within the workflow.
Similar to many radiomics studies, this study is limited by the small number of data samples. Radiomics create thousands of features, meanwhile, the modeling, especially the neural network, uses many parameters. All these require a great number of data samples for training and testing. data samples 143/60 may be not enough.
Indeed, the sample size is one of the limitations of our study, as we stated in the Discussion section (line 404). Being aware of this problem, we have attempted to use the best strategies in data analysis. Future multi-center studies that gather a large number of cases may overcome this issue.
Another common limitation for radiomics is the potential lack of reproducibility and reliability of radiomics feature calculations. While this could be out of scope of this study, the study involves MR from multiple sources. Could the authors evaluate the differential power of a feature between different data sources, and discuss the consistency of the feature predictive power across all the data sources.
Indeed, the study of the variability of the calculation of the radiomic features from the different sources is a subject that goes beyond the objectives of our work. Initially, we had considered the possibility of comparing the results of the classification model and the stability of the set of features used, using each data set (public/institutional) as a test group separately. The disadvantage of performing this analysis is that each source will include some cases used in the training data set. Therefore the results will be biased. For this reason, we finally decided not to include this analysis in the manuscript. Despite this, we send you attached an extra table to see the performance of the best model (Information Gain + Naive Bayes) in each data set. We have grouped the institutional cases because the scanner used is the same and also because, in one of the institutions, there are no cases of survival less than six months.
Another critical limitation that we must mention is that evaluating the reproducibility of the features in the public data set such as the BraTS is not feasible since the data comes from 19 different institutions without knowing which cases correspond to a specific MRI protocol and manufacturer (line 113).
We believe that a specially designed study is necessary to evaluate the reproducibility of the characteristics. Therefore, following your suggestion, we have added this issue in the Discussion section (line 409).

Reviewer 3 Report
Thank you for the opportunity to review this manuscript.
Here, the authors provide a retrospective, two-center study over 2 years years comparing radiomic determinants in glioblastoma patients.
The study is original, well written, brief, and concise. It offers updated information for oncologists, neurosurgeons, and radiotherapists alike.
This technique is promising and after further validation might hold an intrinsic value for treatment algorithms. Furthermore, more subgroups could be identified. All in all this can serve as a building block on the road for personalized medicine.
Layout and format: The manuscript is structured and meets the expected format of the targeted journal.
Title: The title of the manuscript is reflecting the content of the article.
Abstract: The abstract is well-structured and reflects the content of the article.
Introduction: The introduction describes the aim of the study accurately.
Methods and statistics: The authors describe data acquisition and the experimental design. The used statistical tests are appropriate and sufficient.
Results: The presentation of the results is clear and stringent. The significant limitations are listed.
Author Response
We appreciate your comments. Through the use of open-source and user-friendly tools, we hope that our work will extend these image analysis techniques to other centers and involve clinicians in developing new predictive models.
Round 2
Reviewer 2 Report
This quick response clarifies my comments. The small dataset, undecided robustness of radiomics feature extraction, limit the study. With that, this is a well-performed study, the paper is ready for publication.